# Essential Oils and Neuropathic Pain

**DOI:** 10.3390/plants11141797

**Published:** 2022-07-07

**Authors:** Imane Ridouh, Kevin V. Hackshaw

**Affiliations:** 1Dell Medical School, University of Texas, 1601 Trinity St., Austin, TX 78712, USA; imane@utexas.edu; 2Department of Internal Medicine, Division of Rheumatology, Dell Medical School, University of Texas, 1601 Trinity St., Austin, TX 78712, USA

**Keywords:** neuropathic pain, essential oils, lavender, bergamot, billy goat weed, nutmeg, rosemary, eucalyptus, chronic pain

## Abstract

Neuropathic pain is one of the most prominent chronic pain syndromes, affecting almost 10% of the United States population. While there are a variety of established pharmacologic and non-pharmacologic treatment options, including tricyclic antidepressants (TCAs), serotonin-noradrenaline reuptake inhibitors, anticonvulsants, trigger point injections, and spinal cord stimulators, many patients continue to have chronic pain or suboptimal symptom control. This has led to an increased interest in alternative solutions for neuropathic pain such as nutritional supplements and essential oils. In this review, we explore the literature on the most commonly cited essential oils, including lavender, bergamot, rosemary, nutmeg, Billy goat weed, and eucalyptus. However, the literature is limited and largely comprised of preclinical animal models and a few experimental studies, some of which were poorly designed and did not clearly isolate the effects of the essential oil treatment. Additionally, no standardized method of dosing or route of administration has been established. Further randomized control studies isolating the active components of various essential oils are needed to provide conclusive evidence on the use of essential oils for neuropathic pain. In this review, we explore the basis behind some of the essential oils of interest to patients with neuropathic pain seen in rheumatology clinics.

## 1. Introduction

### 1.1. Neuropathic Pain

#### 1.1.1. Background: Definition, Etiology, and Pathogenesis

A consensus definition of neuropathic pain is ‘pain resulting from a lesion of the somatosensory system, which results in faulty pain signaling’ [1]. The somatosensory system includes the peripheral nerves, spinal cord, and the cerebral cortex. Neuropathic pain is associated with many conditions including spinal cord compression, HIV, amputation, fibromyalgia, multiple sclerosis, postherpetic neuralgia, and diabetic neuropathy, and its pathogenesis in each of these states depends on the associated disease process. In the case of diabetic neuropathy, for example, longstanding hyperglycemia induces many metabolic processes including generation of free radicals and deposition of advanced glycation end-products in the microvasculature supplying peripheral nerves. These degradative processes result in direct toxicity and decreased blood flow, leading to nerve fiber degeneration and hyperexcitability of the primary afferent nociceptors [2].

Pain transmission through nociceptive nerve fibers involves many signaling molecules and ion channels, the most well studied of which are briefly mentioned below. Voltage gated sodium channels are involved in neuronal depolarization, and blockage of these channels is the mechanism of action of many first line pain medications. Transient potential receptor channels are expressed on nociceptive nerve fibers, such as C-fibers, and are involved in the development and maintenance of chronic pain. Blockage of voltage gated calcium channels, an alternative conduit involved in neuronal excitability, is the mechanism of action of gabapentin. Glutamate is the main excitatory neurotransmitter released by nociceptive afferent neurons and binds to the N-methyl-D-aspartic acid receptor, which is involved in central sensitization of spinal nociceptive neurons. Conversely, G-Amino Butyric Acid (GABA) is the main inhibitory neurotransmitter, and activation of GABA receptors causes inhibition of signal transmission. Substance P and Calcitonin Gene Related Peptide (CGRP) are neuropeptides released by nociceptive C-fibers and play a role in pain perception as well as signaling in the hypothalamus and amygdala. Finally, opioid receptors are widely distributed through the CNS and periphery. Presynaptically, opioids inhibit neurotransmitter release by reducing Ca2+ influx. Postsynaptically, opioids cause K+ efflux, which hyperpolarizes the cell and decreases the synaptic transmission [3].

Patients experiencing neuropathic pain often have comorbid mood disorders, including depression and anxiety. A 2008 epidemiological study by Gustorff et al. suggests that about 34% of neuropathic pain patients experience feelings of depression, 25% report feelings of anxiety, and 60% report strong or predominant sleep disturbances [4]. Animal models have been used to demonstrate an association between neuropathic pain and depressive behavior with associated changes in the amygdala. Gonclaves et al. found increased signs of depressive-like behavior in rats who had undergone spared nerve injury (SNI), a procedure often used to induce neuropathic pain in animal models. They also noted increase amygdala volume in the SNI rats compared to controls [5]. The amygdala is involved in processing emotional states, and thus may be responsible for the connection between chronic pain and negative emotional states like depression and anxiety. Finally, chronic neuropathic pain causes sleep disruption, but sleep disruption also enhances pain perception and reduces pain tolerance [6].

Taken together, the etiology of neuropathic pain is ultimately multifactorial, as there are well established connections between neuropathic pain, depression, anxiety, and sleep problems.

#### 1.1.2. Impact

The prevalence of neuropathic pain in the United States in a 2009 study was estimated to be about 9.8% [7]. Neuropathic pain management is extremely burdensome on our medical system. The economic burden includes both direct costs related to diagnosing and treating the pain and associated complications, as well as indirect costs due to lost wages, lost productivity, need for home care, and leaving the workforce due to disability [8] A 2004 study found a threefold increase in health care costs in patients with peripheral neuropathy compared to matched controls [9]. Schaefer and colleagues quantified the cost of direct and indirect costs of neuropathic pain in US patients and found mean direct costs to payers to be $6016 and mean indirect cost to be $19,000 per patient. These costs varied significantly but were positively correlated with pain severity [10]. They also found high neuropathic pain levels to be associated with significant decrease in quality of life, including increased anxiety and depression, and worsened sleep status [11]. The prevalence and economic burden of neuropathic pain underscore the importance of exploring new and adjunct treatments.

#### 1.1.3. Focus on Natural Remedies

The focus in treating neuropathic pain is management of symptoms. Neuropathic pain is seen as a chronic condition, and even reversal of the underlying condition does not always result in resolution of the patient’s symptoms. A multimodal approach to management is often necessitated in such individuals, including both pharmacological and nonpharmacological options. Whereas typical pharmacologic interventions include voltage gated calcium channel blockers, serotonin and norepinephrine reuptake inhibitors, sodium channel inhibitors amongst others, a growing portfolio of non-pharmacological therapies have become increasingly utilized for neuropathic pain in recent years with varying degrees of efficacy. These include massage therapy and acupuncture, TENS units, trigger point injections, and spinal cord stimulators. With increased consumer interest in “natural” alternative options, essential oils, herbal remedies, and dietary supplements have gained significant traction as possible adjunct treatment options. This interest is particularly high in regions of the world with a long history of traditional medicine, including large parts of Asia, Africa, and Latin America [12].

## 2. Essential Oils

### 2.1. Definition

Essential oils are defined as the volatile compounds synthesized by any part of the plant, including leaves, stems, or flowers. These compounds provide the unique flavor and scent of the plant and stored in the form of hydrophobic lipid soluble droplets in the plant cell wall, secretory cells, glandular hairs, and epidermal cells. These essential oils represent the byproducts of the plant’s metabolic processes and are made up of volatile and nonvolatile fractions collected through a distillation process [13]. The volatile fractions are comprised of mono- and sesquiterpene components, their oxygenated derivatives, alcohols, aliphatic aldehydes, and esters, while nonvolatile fractions are derived from carotenoids, flavonoids, fatty acids, and waxes [14].

### 2.2. Purported Uses

Traditional eastern medicine has a longstanding history of using essential oils for a wide variety of health aspects. Lavender is used as aromatherapy for anxiety, sleeplessness, and headaches in Chinese medicine. Bergamot is used for indigestion and nausea. Clove oil is used in ayurvedic medicine—another traditional approach which shares many similarities with traditional Chinese medicine—as an immune boosting chemo preventative agent [15]. Furthermore, there are scant amounts of literature studying the biological activity of many essential oils, including anticarcinogenic, antioxidant, anti-inflammatory, antifungal, and antibacterial activities [16]. For example, cardamom essential oil was found to have antibacterial effect against Pseudomonas Aeruginosa and Escherichia Coli as well as an antidiarrheal and anti-spasmodic effect [17]. Inhalation of patchouli essential oil decreased low density lipoprotein (LDL) and systolic blood pressure (BP) in rats [18]. Eucalyptus essential oil has antiviral activity against herpes simplex virus {HSV)-1, HSV-2, influenza A, coxsackie, and rotavirus [19]. Lavender essential oil has anti-inflammatory properties and work at the cellular level by decreasing pro-inflammatory cytokine mRNA expression and IL-8 secretion [20]. In studies by Takikawa et al., nutmeg essential oil had antibacterial effects on pathogenic strains of Escherichia coli, and inhibited Yersinia entero-colitica and Listeria monocytogenes growth in broth culture [21].

### 2.3. Global Market

As of 2020, the global essential oil industry is valued at 18.6 billion US dollars and expected to grow 7.5% compounded annually over the next eight years, reaching an es-timated 35.5 billion by 2028. Orange oil occupied the largest share at 9.5% of global essential oil revenue in 2020, with davana oil predicted to show the highest compound annual growth rate of 16.9% due to its use in natural perfumes and various medical applications. Although the spa and relaxation sector take up about 46% of the essential oil market, market analysis revealed the increasing popularity of integrative medicine as one of the main drivers for increase in essential oil demand. Finally, Europe currently leads the market with 49.2% of global revenue, which is attributed to in-creased consumer awareness of essential oils. The market is expected to continue to grow in Asia Pacific, Africa, and the Middle East due to local demand and shifts in consumer preferences towards healthy lifestyles [22].

### 2.4. Lack of Evidence

There is a wide variety in how these oils are used. Some are topically applied to areas of neuropathic pain while others are inhaled as aromatherapy. Some studies isolate the active ingredient whereas others use herbal extracts. There are no clear dosage guidelines and the degree to which studies dilute essential oils in a carrier oil varies dramatically.

This paper details the current literature on the use of essential oils for neuropathic pain. Articles were sourced by using any combination of the key words “essential oil”, “neuropathic pain”, “neuropathy”, “chronic pain”, and “chronic widespread pain”. We used a robust biomedical database, PubMed, which resulted in 39 hits, 35 of which were published between 2010 and 2022. Use of the Google Scholar search engine resulted in 23,000 results, which included all articles that had the search terms anywhere in the text. A joint review of these results by the authors helped narrow the search to the most relevant studies. Since the focus of our interest was on neuropathic pain remedies, these two databases were deemed to be sufficient for the purposes of our literature search. In addition, since 90% of relevant citations were noted between 2010 and 2022, we decided to focus our review to this time span. The most studied essential oils that are seen in a university neuropathic pain practice and also noted in literature are summarized below.

## 3. Lavender (Lavendula) for Neuropathic Pain

Most commonly sourced from the flower of the Lavandula angustifolia species, lavender essential oil has been used historically to alleviate stress and anxiety. Lavender aromatherapy massage and essential oil inhalation have been documented to provide pain relief in a wide variety of states, which includes pain from dysmenorrhea, osteoarthritis, and labor [23].

A 2017 study by Metin et al. examined diabetic patients with neuropathic pain at an outpatient clinic in Turkey. The experimental group received a massage oil of equal parts lavender, rosemary, geranium, and eucalyptus diluted to 5% in coconut oil. Patients received 20-min foot massages and 10-min hand massages three times a week for four weeks. Findings showed significantly decreased levels of neuropathic pain and increased quality of life (QoL) metrics [24]. Of note, this study was poorly designed as the control group did not have any interventions. It is unclear if the reduction in neuropathic pain was a result of massage therapy or a specific essential oil found in the aromatherapy blend (Table 1). However, a 2021 randomized control trial (RCT) by Mozhgan et. al. replicated similar findings. They looked at both pain scores and QoL metrics for diabetic neuropathy patients randomized to receive 10 min of foot massage with 10% lavender oil in sunflower oil for one month at outpatient clinics in Iran. They found a significant increase in QoL metrics and a decrease in neuropathic pain at the four-week mark [25].

In a study using mice with spared nerve injury to model neuropathic pain, pure lavender (L. angustifolia) essential oil was diluted in 5% DMSO and administered orally 30 min before tests performed 7 days after the nerve injury. The effects of a single oral dose of lavender essential oil were comparable to that of pregabalin, a pharmacological agent often prescribed for neuropathic pain. It increased the mice’s pain threshold and reversed mechanical hypersensitivity. Measurement of spinal cord levels of various cellular pathway intermediaries implicated in neuropathic pain revealed oral lavender essential oil led to a marked inhibition of spinal extracellular signaling-regulated kinase (ERK) and c-Jun N-terminal kinase (JNK) phosphorylation and reduction of nitric oxide synthase (iNOS) ex-pression [26]. ERK is activated in the dorsal spinal cord by both nociceptive stimulation as well as nerve damage. Inhibiting ERK signaling reduces behavioral response to nociceptive stimulus. This suggests that ERK activation is involved in nociceptive signaling [26]. Thus, reduction in phosphorylation of ERK and other intermediaries may be one mechanism by which it decreases pain signaling.

The mechanism of linalool (Table 2), a component of lavender essential oil, has also been studied in isolation. Peana et al. has conducted studies demonstrating cholinergic, local anesthetic and NMDA receptor blockade. Moreover, these studies imply that mechanisms may be related to the opening of potassium (K+) channels, which may occur due to stimulation of muscarinic M2, opioid, or dopamine D2 receptors [27].

Furthermore, Tashiro et al. has shown on orexin neuron-deficient and orexin pep-tide-deficient mice exposed to formalin tests that orexinergic transmission was essential for linalool odor-induced analgesia, signifying that linalool caused the activation of hypothalamic orexin neurons, which act as key mediators for processing pain [28].

**Table 2 plants-11-01797-t002:** Essential Oil, Key volatile components, chemical structure and Plant.

Essential Oil	KEY Volatile Component(s)	Chemical Structure	Plant
LAVENDER	Linalyl acetate **LINALOOL** Tannins Caryophyllene	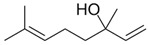	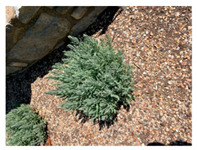
BERGAMOT	D-Limonene **LINALYL ACETATE** Linalool α-Terpinene/β- Pinene	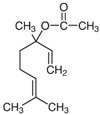	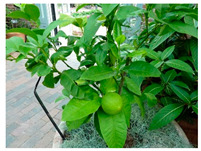 [29]
NUTMEG	**TERPENE****HYDROCARBONS** Oxygenated Terpenes (**LINALOOL**) Aromatic ethers	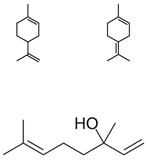	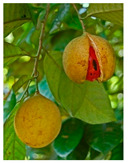 [30]
ROSEMARY	**1,8 CINEOLE** Camphor α−Pinene	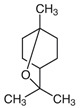	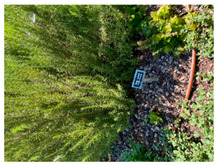
BILLY GOAT WEED	Terpinine-4-ol Bornyl acetate E-Caryophyllene γ -Muroleno δ -Cadinene α-Muroleno Caryophyllene-oxide **LONGIFOLENE** α-Humulene Precocene I & II	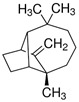	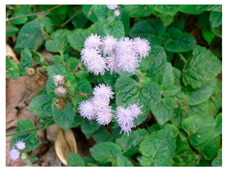 [31]
EUCALYPTUS	**1,8 CINEOL** α-Pinene	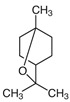	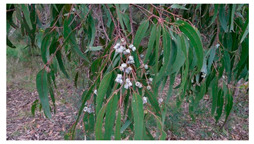 [32]

While lavender essential oil from the flower of the Lavandula angustifolia is cur-rently the most commonly available commercially, the oils of many other species are available including L. spica, L. spicata, L. stoechas, L. x intermedia, L. officinalis, L. vera, and L. latifolia. These have gained popularity in online communities at a significantly faster rate than research examining their effects and safety and present an opportunity for further research to better understand the significance of oils sources from different species.

## 4. Bergamot (*Citrus bergamia*) for Neuropathic Pain

Bergamot essential oil (BEO) is derived from the rind of the bergamot orange fruit and shares many of the same volatile compounds as lavender. BEO used in the research by Komatsu et al. [33] was composed 0.38% d-limonene, 70.26% linalyl acetate, 18.95% linalool, 0.62% γ-terpinene, and 0.03% of β-pinene, although specific compositions vary widely based on where the oil is sourced from and how it is prepared.

Research by the Tsukasa Sakurada group based out of Japan had previously established that the nociceptive responses in mice exposed to nociceptive stimuli in the form of capsaicin or formalin was inhibited by BEO injection [34]. This was reversed with administration of naloxone hydrochloride, which is an opioid receptor antagonist [35]. This suggests that BEO’s antinociceptive effect is mediated in part through the opioid receptor. Building upon this work, they examined the role of BEO specifically in neuropathic pain using mice with partial sciatic nerve ligation. BEO was injected at three different doses into the hind paw of the mouse (5.0, 10.0, and 20.0 μg/paw) and was found to dose-dependently attenuate partial sciatic nerve ligation induced allodynia compared to jojoba wax controls. Furthermore, injection into the contralateral leg of the sciatic nerve ligation did not show any antiallodynic effect of BEO injection, suggesting that the mechanism of attenuating neuropathic pain is local rather than systemic [36].

In trying to better understand the mechanism of action, the Sakurada group found that pretreating with a selective μ-opioid receptor antagonist reversed the BEO effects seen, but pretreatment treatment with a nonselective δ -opioid receptor antagonist and a selective κ-opioid receptor antagonist did not have any effect on the antiallodynic effect of BEO. This suggests that BEO’s effects are tied to the μ-opioid receptor. They hypothesize that BEO may cause the release of endogenous opioids, or may directly activate the opioid receptor. Finally, ERK phosphorylation levels in the spinal cord were found to be decreased with BEO administration [36]. This mirrors the findings of phosphorylated ERK levels decreasing with lavender essential oil administration detailed above.

## 5. Nutmeg (*Myristica fragrans*) for Neuropathic Pain

Nutmeg essential oil comes from the seed of the *Myristica fragrans* Houtt plant, and its major components are terpene hydrocarbons which make up 60% to 80% of the oil, oxygenated terpenes, like linalool, which make up approximately 5% to 15%, and aromatic ethers making up the remaining 15 to 20% [37].

There is currently only one study analyzing the anti-neuropathic effect of nutmeg essential oil for painful diabetic neuropathy, in which the authors concluded there was no evidence to support any change in symptoms over 4 weeks of treatment with nutmeg essential oil. In this study, 74 patients with painful diabetic neuropathy in a hospital in Trinidad were randomized to receive either a topical nutmeg extract in a menthol coconut oil base, or solely the menthol coconut oil base. Patients sprayed the solution on with a spray bottle to the affected area 3 times a day for four weeks followed by a gentle massage. Results were measured with two measures by a blinded assessor—the Brief Pain Inventory validated for Diabetic Painful Neuropathy (BPI-DPN) and Neuropathic Pain Symptom Inventory (NPSI). These assessed pain, sleep, and functional ability. There were no statistically significant differences between the groups at the end of the follow-up period for any outcome measure. This was a well-designed study, however using menthol as part of the carrier oil for both the experimental and control groups may mask the effect of nutmeg oil. Menthol is an active ingredient itself and may have been providing some anti-allodynic effect in the diabetic neuropathy patients, such that nutmeg essential oil did not provide any further benefit [38].

## 6. Rosemary (*Salvia rosmarinus*) for Neuropathic Pain

Rosemary essential oil is sourced from the leaves of Rosmarinus officinalis L. and consists mostly of monoterpenes such as 1,8-cineole, camphor and α-pinene [39]. The data on rosemary essential oil is limited, although it is one of the more popular homeopathic essential oils for various medical ailments including renal colic and dysmenorrhea. There are some studies looking at rosemary essential oil’s analgesic and anti-nociceptive effect in animal models, although none specifically looking at neuropathic pain models. For example, a study by Mikov et al. looked at the effect of orally administered rosemary essential oil in combination with codeine and acetaminophen on latency to withdrawal to heat induced pain in mice. They found a dose dependent analgesic effect compared to saline control. This suggests that rosemary essential oil would be a promising direction for further research [40].

## 7. Billy Goat Weed (*Ageratum conyzoides*) for Neuropathic Pain

Billy goat weed, also known as *Ageratum conyzoides*, is an invasive weed in many tropical countries. Its major volatile components are terpinene-4-ol, bornyl acetate, E-caryophyllene, γ-muroleno, δ-cadinene, α-muroleno, caryophyllene oxide, longifolene, α-humulene, and precocene I and II. [40] There is only one study in the literature examining billy goat weed essential oil for neuropathic pain. Based out of Indonesia, Adenyana et al. tested steam distilled essential oil against nonessential oil components (steam distillation residue) as well as negative control, pregabalin, and naloxone in chronic constriction injury mice as models for neuropathic pain. Of note, they do not clarify in their manuscript the method of essential oil administration. However, their results show that the mice in the essential oil group had equal responses to the pregabalin mice in both hyperalgesia and allodynia tests on a hotplate. They demonstrated less response to pain compared to the sham group and the non-essential oil component groups. Administration of naloxone, an opioid receptor antagonist, abolished the anti-neuropathic pain effect in the essential oil and pregabalin groups. This suggests the opioid receptor is involved in the analgesic effect of billy goat weed essential oil [41]. This is similar to the effect of bergamot essential oil described previously and may suggest that a shared essential oil active ingredient is involved in opioid receptor signaling.

A follow up study by Adenyana et al. isolated the three main *Ageratum conyzoides* essential oil components: precocene II, longifolene, and caryophyllene. Using chronic constriction injury rat models for neuropathic pain, they found that administration of caryophyllene and longifolene has anti-neuropathic pain activity, while percocene did not. Method of administration was unclear. Both longifolene and caryophyllene partially had an effect through opioid receptor activation, while caryophyllene also resulted in increased GABA levels in the spinal cord. Interestingly the combination of pregabalin with *Ageratum conyzoides* essential oil, longifolene, or caryophyllene had a synergistic effect in anti-neuropathic pain activity [42].

## 8. Eucalyptus (*Eucalyptus globulus*) for Neuropathic Pain

Eucalyptus essential oil is made up of primarily two volatile components: 1,8-cineol (49.07 to 83.59%) and α-pinene (1.27 to 26.35%) [43]. Liu et al. isolated 1,8-cineol and specifically studied its effect on purinoreceptor 3 (P2X3) receptor-mediated neuropathic pain, followed by a publication on the effect on the P2X2 receptor a year later. P2X3 receptors are located on small to medium diameter C and Aδ afferent neurons, which carry mechanical, chemical, and thermal pain information. Thus, antagonism of the P2X3 receptors is a potential therapeutic strategy for pain relief [44]. The first study used a rat model of chronic constriction injury (CCI) to represent neuropathic pain and gave the experimental group daily intragastric doses of 1,8-cineol, with both non CCI controls and DMSO intragastric dose controls. Rats receiving 1,8-cineol showed significant decreases in various behavioral measures of neuropathic pain, including mechanical paw withdrawal threshold and thermal withdrawal latency values, which suggests that 1,8-cineol alleviates the allodynia. Furthermore, P2X3 mRNA PCR, immunohistochemical staining, and western blot of the dorsal root ganglia showed a downregulation in P2X3 receptors in rats receiving 1,8-cineol. All rats that had received a CCI showed an upregulation of P2X3 ipsilateral to the CCI, suggesting that upregulation of the receptor plays a role in neuropathic pain transmission, and this is countered by 1,8-cineol. Thus, this study suggests a possible mechanism by which 1,8-cineol alleviates neuropathic pain: it inhibits P2X3 receptor mRNA transcription and translation to inhibit pain signal transmission. Interestingly, the paper notes that route of administration is an area for further research, because it is unclear whether 1,8-cineol would be fully functional after oral administration. Furthermore, there is limited data about safety and dosage limits, and a preliminary experiment of intraperitoneal injection of 1,8-cineole caused the death of some rats [45].

Their following study applied much of the same methodology to examine the effect of intragastric 1,8-cineol administration on the P2X2 receptor in neuropathic pain animal models. The P2X2 receptor is closely related to the P2X3 receptor and also implicated in transmission of analgesia. Oral administration of 1,8-cineol decreased P2X2 receptor mRNA and protein levels in the spinal cord [46].

Together, these studies suggest the specific eucalyptus volatile isolate, 1,8-cineol, may have an anti-allodynic effect on neuropathic pain.

## 9. Summary

In summary, there is limited evidence to support the use of essential oils for neu-ropathic pain, and much of the existing research is based on animal models. There is also very little standardization in essential oil dosage or even method of administration. Furthermore, lack of FDA regulation means there is variability in how the oils are sourced and their resulting concentration of active components. Focusing in on active ingredients may be an area of promising further research. This will allow precise dosage and standardization across studies.

The essential oil market is rapidly growing and will likely continue to be explored as adjunct therapy for various pain syndromes. There is enough evidence to suggest that some essential oil active components have direct anti-neuropathic activity and modulate pain signaling. However, there is likely an aspect of emotional relief that results in patients feeling better and adhering to essential oil use. The interplay between neuropathic pain, depression, and anxiety is bidirectional, and improving a patient’s emotional state may lead to decrease pain sensitivity. Essential oils have been studied to a greater degree for depression and anxiety prevention and treatment, and work through a variety of mechanisms including monoamine level regulation and promotion of neurogenesis [47]. More research controlling for symptoms of depression and anxiety is needed to tease apart how much of the effects of various essential oils are due to direct modulation of pain signaling and how much is due to improvement of mood symptoms.

This review represents a survey of the most frequently encountered essential oils used by patients seen in a university pain practice. Drawbacks of this type of review are that it is by no means exhaustive as the oils discussed represent those most frequently obtained in the North American marketplace. Multiple new essential oil compounds are being discovered and efficacy evaluated globally [48,49,50]. Thus, this review suffers from having a continental rather than global outlook as most of the essential oils discussed are frequently used by individuals in the United States. As many of the newer agents being identified globally gain traction in U.S. markets, these newly discovered oils may gain increased favor here and become more widely used.

While there is no extensive evidence supporting the use of essential oils for neuropathic pain, inhalation and topical application is generally safe and has low rates of serious adverse effects. Thus, there is not enough evidence to discourage patients who are interested in trying essential oil therapies, although they should be counseled on obtaining it through reputable sources and diluting in a carrier oil before use.

## Figures and Tables

**Table 1 plants-11-01797-t001:** Essential Oil Controlled Neuropathic Pain Studies.

Essential oil	Human Studies	Significance	Comments	Animal Studies	Significance	Comments
LAVENDER	Y	Y	PD	Y	Y	Mice
BERGAMOT	N	-	-	Y	Y	
NUTMEG	Y	NS	74 subj (DPN)	N	-	-
ROSEMARY	N	-	-	N	-	-
BILLY GOAT WEED	N	-	-	Y	+/−	Rat (Poorly described)
EUCALYPTUS	N	-	-	Y	Y	Rat

PD = Poor design. DPN = Diabetic peripheral neuropathy. NS = Not significant. Y = Yes. N = No.

## Data Availability

Not applicable.

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
