# Peer review of "Essential Oils and Neuropathic Pain"

_plants, 2022, doi:10.3390/plants11141797_

Round 1

Reviewer 1 Report

In this review, the authors discuss the potential role of essential oils in treating neuropathic pain. The authors conducted a literature search and found that there is limited evidence to support the use of essential oils for neuropathic pain, and most studies use preclinical animal models.  Nevertheless, this is a well-written and easy to follow review that highlights the increasing need for further research in this area and the ever-expanding market for essential oil products.

Many words throughout the manuscript are unnecessarily hyphenated when they do not need to be. Most likely, this is a formatting error, but it should be corrected.

Section 1.1.2 Impact: what does BEAT stand for? ‘The 2014 BEAT study’

Section 2.1: change ‘mono and sesquiterpene components’ to ‘mono- and sesquiterpene components’

Section 3: change ‘The mechanism of linlool,’ to ‘The mechanism of linalool,’

Section 4: is the entire section based on reference 30 or are you referring to several different references? ‘Research by the Tsukasa Sakurada group based out of Japan…’

Section 9: change ‘improving a patient emotional state’ to ‘improving a patient’s emotional state’

Author Response

Thank you for your review and comments. A point by point response to the concerns is listed below. 

  1. A spell check has been conducted and unnecessary hyphenations have been removed.
  2. Referral to the 2014 Beat study has been removed. It is now referred to as "Schaefer and colleagues..." Track changes notes the changes. 
  3. Change has been made as requested:  mono and sesquiterpene has been changed to mono- and sesquiterpene components
  4. Section 3 : correction made to linalool
  5. Section 4: All of this work is from the same group. Two additional references have been added to this section. 
  6. Section 9: Change has been made to improving a patient's emotional state

Reviewer 2 Report

The article presented to me for review raises an important topic, which is the fight against chronic pain, here with the use of essential oils.

The manuscript is well structured, interesting and worth publishing in Plant.

Nevertheless, I have a few comments:

1 / please, specify which oils this article concerns

2 / please add photos of the discussed plants 

3 / it is worth diversifying the work by inserting a graph / table in the part where the values are quoted

4 / please follow the Plant requirements in the References section

Author Response

Thank you for your review and comments. 

Items 1, 2, and 3 have been addressed with inclusion of Table 1 and Figure 1 which includes examples of all the plants. 

Reference section has been updated to comply with Plants journal reference format.

Reviewer 3 Report

The authors should consider the followings:

1.          The authors may use a comprehensive table to summarize the concerned publication, by the stage(s) and type(s) of clinical studies, number of patients, the significance and limitation of the work.

2.          The authors should include a section to list the limitation of the current review.

3.          The authors should use a comprehensive table to summarize the concerned publication, by the animal models, number of animal used, the significance and limitation of the work.

4.          The author may use tables to present the "lack of evidence" with the respective publication.

5.          As a review, the authors should include more relevant references to this review article.

6.          The authors should give rationales of limiting the publication year between 2010 and 2022; where the search in Google scholar also limited to publication year between 2010 and 2022.

7.          Across the article, the authors should give full name (before assigning the abbreviation), such as in the section 2.2, BP, HSV and LDL.

8.          In the respective paragraph, the authors should give the correct botanical naming of the plants accordingly, lavender, bergamot, rosemary, nutmeg, billy goat weed, and eucalyptus.

9.          The authors may use one figure to show the chemical structures of the active ingredients described across the review.

10.      The author can use tables to show the citation scores of those essential oils (with the respective criteria)

11.      Please correct the hypenated words, al-most, and trig-ger (line 9 and line 12, of page 1), as well as other concerned hypenated words in the article.

12.      The author should define the words, most commonly cited essential oils.

13.      Some comprehensive workflow or tables should be used in this review to let a greater audience to understand this review.

14.      The authors should explain the use of only Pubmed and google scholar but not other databases.

Author Response

Thank you for your review and comments. See specific responses to items below.

  1. The authors may use a comprehensive table to summarize the concerned publication, by the stage(s) and type(s) of clinical studies, number of patients, the significance and limitation of the work.

Table 2 has been added which includes this information

  1. The authors should include a section to list the limitation of the current review.

New paragraph, lines 376 – 384 address the limitations.

  1. The authors should use a comprehensive table to summarize the concerned publication, by the animal models, number of animal used, the significance and limitation of the work.

Table 2 has been added which includes this information

  1. The author may use tables to present the "lack of evidence" with the respective publication.

Table 2 has been added which includes this information

  1. As a review, the authors should include more relevant references to this review article.

Additional references have been added to this review.

  1. The authors should give rationales of limiting the publication year between 2010 and 2022; where the search in Google scholar also limited to publication year between 2010 and 2022.

See lines 153-163 which further explains the rationale for limitation of publication year between 2010 and 2022

  1. Across the article, the authors should give full name (before assigning the abbreviation), such as in the section 2.2, BP, HSV and LDL.

This has been corrected.

  1. In the respective paragraph, the authors should give the correct botanical naming of the plants accordingly, lavender, bergamot, rosemary, nutmeg, billy goat weed, and eucalyptus.

This has been corrected

  1. The authors may use one figure to show the chemical structures of the active ingredients described across the review.

Chemical structure of the key active ingredients for each species is shown in table 1.In addition, an example of each plant species is shown.

  1. The author can use tables to show the citation scores of those essential oils (with the respective criteria)

Not sure what the reviewer was requesting here.

  1. Please correct the hypenated words, al-most, and trig-ger (line 9 and line 12, of page 1), as well as other concerned hypenated words in the article.

All hyphenated words have been corrected throughout the manuscript.

  1. The author should define the words, most commonly cited essential oils.

See lines 153-163 which further explains the choice of the essential oils that are discussed/reviewed

  1. Some comprehensive workflow or tables should be used in this review to let a greater audience to understand this review.

Tables 1 and 2 provide clarity for a greater audience to understand this review.

  1. The authors should explain the use of only Pubmed and google scholar but not other databases.

Line 153 – 163 discuss the reasons for use of only Pubmed and google scholar.